# Transfer-Free Data-Efficient Multilingual Slot Labeling

**Evgeniia Razumovskaia    Ivan Vulić    Anna Korhonen**
Language Technology Lab, University of Cambridge
{er563, iv250, alk23}@cam.ac.uk

## Abstract

Slot labeling (SL) is a core component of task-oriented dialogue (ToD) systems, where slots and corresponding values are usually language-, task- and domain-specific. Therefore, extending the system to any new language-domain-task configuration requires (re)running an expensive and resource-intensive data annotation process. To mitigate the inherent data scarcity issue, current research on multilingual ToD assumes that sufficient English-language annotated data are *always* available for particular tasks and domains, and thus operates in a standard cross-lingual transfer setup. In this work, we depart from this often unrealistic assumption. We examine challenging scenarios where such transfer-enabling English annotated data cannot be guaranteed, and focus on bootstrapping multilingual data-efficient slot labelers in *transfer-free* scenarios directly in the target languages without any English-ready data. We propose a **two**-stage **s**lot **l**abeling approach (termed TWOSL) which transforms standard multilingual sentence encoders into effective slot labelers. In Stage 1, relying on SL-adapted contrastive learning with only a handful of SL-annotated examples, we turn sentence encoders into task-specific span encoders. In Stage 2, we recast SL from a token classification into a simpler, less data-intensive span classification task. Our results on two standard multilingual ToD datasets and across diverse languages confirm the effectiveness and robustness of TWOSL. It is especially effective for the most challenging transfer-free few-shot setups, paving the way for quick and data-efficient bootstrapping of multilingual slot labelers for ToD.

## 1 Introduction and Motivation

Slot labeling (SL) is a crucial natural language understanding (NLU) component for task-oriented dialogue (ToD) systems (Tur and De Mori, 2011). It aims to identify slot values in a user utterance and fill the slots with the identified values. For instance, given the user utterance *"Tickets from Chicago to Milan for tomorrow"*, the airline booking system should match the values *"Chicago"*, *"Milan"*, and *"tomorrow"* with the slots departure_city, arrival_city, and date, respectively.

Building ToD systems which support new domains, tasks, and also languages is challenging, expensive and time-consuming: it requires large annotated datasets for model training and development, where such data are scarce for many domains, tasks, and most importantly - languages (Razumovskaia et al., 2022a). The current approach to mitigate the issue is the *standard cross-lingual transfer*. The main 'transfer' assumption is that a suitable large English annotated dataset is always available for a particular task and domain: (i) the systems are then trained on the English data and then directly deployed to the target language (i.e., zero-shot transfer), or (ii) further adapted to the target language relying on a small set of target language examples (Xu et al., 2020; Razumovskaia et al., 2022b) which are combined with the large English dataset (i.e., few-shot transfer). However, this assumption might often be unrealistic in the context of ToD due to a large number of potential tasks and domains that should be supported by ToD systems (Casanueva et al., 2022). Furthermore, the standard assumption implicitly grounds any progress of ToD in other languages to the English language, hindering any system construction initiatives focused directly on the target languages (Ruder et al., 2022).

Therefore, in this work we depart from this often unrealistic assumption, and propose to focus on *transfer-free scenarios* for SL instead. Here, the system should learn the task in a particular domain *directly* from limited resources in the target language, assuming that any English data cannot be guaranteed. This setup naturally calls for constructing a versatile multilingual *data-efficient* method that leverages scarce annotated data as effectively as possible and should thus be especially applicable to low-resource languages (Joshi et al., 2020).

Putting this challenging setup into focus, we thus propose a novel **two**-stage **s**lot-**l**abeling approach, dubbed TWOSL. TWOSL recasts the SL task into a span classification task within its two respective stages. In Stage 1, a multilingual general-purpose sentence encoder is fine-tuned via contrastive learning (CL), tailoring the CL objective towards SL-based span classification; the main assumption is that representations of phrases with the same slot type should obtain similar representations in the specialised encoder space. CL allows for a more efficient use of scarce training resources (Fang et al., 2020; Su et al., 2021; Rethmeier and Augenstein, 2021). Foreshadowing, it manages to separate the now-specialised SL-based encoder space into slot-type specialised subspaces, as illustrated later in Figure 2. These SL-aware encodings are more interpretable and allow for easier classification into slot types in Stage 2, using simple MLP classifiers.

We evaluate TWOSL in transfer-free scenarios on two standard multilingual SL benchmarks: Multi-ATIS++ (Xu et al., 2020) and xSID (van der Goot et al., 2021), which in combination cover 13 typologically diverse target languages. Our results indicate that TWOSL yields large and consistent improvements **1)** across different languages, **2)** in different training set size setups, and also **3)** with different input multilingual encoders. The gains are especially large in extremely low-resource setups. For instance, on MultiATIS++, with only 200 training examples in the target languages, we observe an improvement in average $F_1$ scores from 49.1 without the use of TWOSL to 66.8 with TWOSL, relying on the same multilingual sentence encoder. Similar gains were observed on xSID, and also with other training set sizes. We also report large gains over fine-tuning XLM-R for SL framed as the standard token classification task (e.g., from 50.6 to 66.8 on MultiATIS++ and from 43.0 to 52.6 on xSID with 200 examples), validating our decision to recast the task in TWOSL as a span classification task.

In summary, the results suggest the benefits of TWOSL for *transfer-free* multilingual slot labeling, especially in the low-resource setups when only several dozen examples are available in the target language: this holds promise to quicken SL development cycles in future work. The results also demonstrate that multilingual sentence encoders can be transformed into effective span encoders using contrastive learning with a handful of examples. The CL procedure in TWOSL exposes their phrase-level semantic 'knowledge' (Liu et al., 2021; Vulić et al., 2022). In general, we hope that this work will inspire and pave the way for further research in the challenging *transfer-free few-shot* setups for multilingual SL as well as for other NLP tasks. The code for TWOSL will be available online.

## 2   Related Work

**Multilingual Slot Labeling.** Recently, the SL task in multilingual contexts has largely benefited from the development of multilingually pretrained language models (PLMs) such as mBERT (Devlin et al., 2019) and XLM-R (Conneau et al., 2020). These models are typically used for zero-shot or few-shot multilingual transfer (Xu et al., 2020; Krone et al., 2020; Cattan et al., 2021). Further, the representational power of the large multilingual PLMs for cross-lingual transfer has been further refined through adversarial training with latent variables (Liu et al., 2019) and multitask training (van der Goot et al., 2021).

Other effective methods for cross-lingual transfer are translation-based, where either the training data in the source language is translated into the target language or the evaluation data is translated into the source (*translate-train* and *translate-test*, respectively; Schuster et al. (2019); Razumovskaia et al. (2022a)). The issues with these methods for SL are twofold. First, the translations might be of lower quality for low-resource languages or any language pair where large parallel datasets are lacking. Second, they involve the crucial *label-projection step*, which aligns the words in the translated utterances with the words in the source language. Therefore, (i) applying translation-based methods to sequence labeling tasks such as SL is not straightforward (Ponti et al., 2021), (ii) it increases the number of potential accumulated errors (Fei et al., 2020), and (iii) requires powerful word alignment tools (Dou and Neubig, 2021).

Several methods were proposed to mitigate the issues arising from the label-projection step. Xu et al. (2020) propose to jointly train slot tagging and alignment algorithms. Gritta and Iacobacci (2021) and Gritta et al. (2022) fine-tune the models for post-alignment, i.e., explicitly aligning the source and translated data for better cross-lingual dialogue NLU. These approaches still rely on the availability of parallel corpora which are not guaranteed for low-resource languages. Thus, alternative approaches using code-switching (Qin et al.,

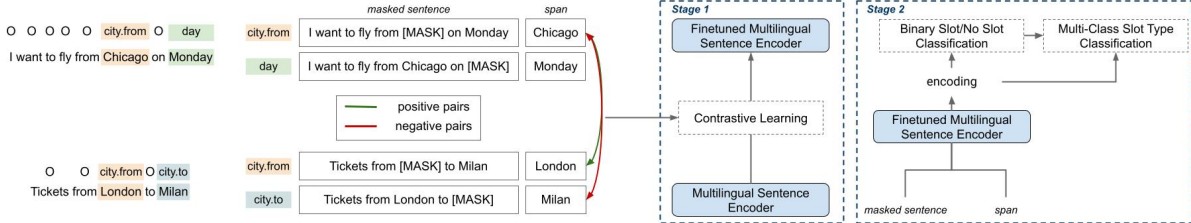

Figure 1: Illustration of the proposed TWOSL framework which turns general-purpose multilingual sentence encoders into efficient slot labelers via two stages. **Stage 1:** contrastive learning tailored towards encoding sub-sentence spans. **Stage 2:** slot classification in two steps, binary slot-span identification/filtering (Step 1, aiming to answer the question *'Is this span a value for any of the slot types?'*) and multi-class span-type classification (Step 2, aiming to answer the question *'What class is this span associated with?'*). Ablation variants include: **a)** using off-the-shelf multilingual sentence encoders in Stage 2 without their CL-based fine-tuning in Stage 1; **b)** directly classifying slot spans without the binary filtering step (i.e., without Step 1 in Stage 2).

2020; Krishnan et al., 2021) were proposed. All of the above methods assume the availability of an 'aid' for cross-lingual transfer such as a translation model or a bilingual lexicon; more importantly, they assume the existence of readily available task-annotated data in the source language.

**Data-Efficient Methods for Slot Labeling.** One approach to improve few-shot generalisation in ToD systems is to pretrain the models in a way that is specifically tailored to conversational tasks. For instance, ConVEx (Henderson and Vulić, 2021) fine-tunes only a subset of decoding layers on conversational data. QANLU (Namazifar et al., 2021) and QASL (Fuisz et al., 2022) use question-answering for data-efficient slot labeling in monolingual English-only setups by answering questions based on reduced training data.

In addition, methods for zero-shot cross-lingual contrastive learning have been developed (Gritta et al., 2022; Gritta and Iacobacci, 2021) which allow for efficient use of available annotated data. Further, Qin et al. (2022) and Liang et al. (2022) use code-switched examples to improve performance on intent classification and slot labeling in a zero-shot setting by training on code-switched examples on slot-value, sentence and word levels. Unlike prior work, TWOSL focuses on adapting multilingual sentence encoders to the SL task in transfer-free low-data setups.

## 3 Methodology

**Preliminaries.** We assume the set of $N_s$ slot types $\mathcal{S} = \{SL_1, \ldots, SL_{N_s}\}$ associated to an SL task. Each word token in the input sentence/sequence $s = w_1, w_2, \ldots, w_n$ should be assigned a slot label $y_i$, where we assume a standard BIO tagging

scheme for sequence labeling (e.g., the labels are $O$, $B\text{-}SL_1$, $I\text{-}SL_1$,..., $I\text{-}SL_{N_s}$).[1] We also assume that $M$ SL-annotated sentences are available in the target language as the only supervision signal.

The full two-stage TWOSL framework is illustrated in Figure 1, and we describe its two stages in what follows.

### 3.1 Stage 1: Contrastive Learning for Span Classification

Stage 1 has been inspired by contrastive learning regimes which were proven especially effective in few-shot setups for cross-domain (Su et al., 2022; Meng et al., 2022; Ujiie et al., 2021) and cross-lingual transfer (Wang et al., 2021; Chen et al., 2022), as well as for task specialisation of general-purpose sentence encoders and PLMs for intent detection (Mehri and Eric, 2021; Vulić et al., 2021). To the best of our knowledge, CL has not been coupled with the ToD SL task before.

**Input Data Format for CL.** First, we need to reformat the input sentences into the format suitable for CL. Given $M$ annotated sentences, we transform each of them into $M$ triples of the following format: $(s_{mask}, sp, L)$. Here, (i) $s_{mask}$ is the original sentence $s$, but with word tokens comprising a particular slot value masked from the sentence; (ii) $sp$ is that slot value span masked from the original sentence; (iii) $L$ is the actual slot type associated with the span $sp$. Note that $L$ can be one of the $N_s$ slot types from the slot set $\mathcal{S}$ or a special None value denoting that $sp$ does not capture a proper slot value. One exam-

---

[1]For simplicity and clarity, we will focus on the standard BIO scheme, while the method is fully operational with other tagging schemes as well.

ple of such a triple is ($s_{mask}$=*Ich benötige einen Flug von [MASK] [MASK] nach Chicago*, $sp$=*New York*, $L$=departure_city). In another example, ($s_{mask}$=*[MASK] mir die Preise von Boston nach Denver, sp=Zeige, L*=None). Note that $sp$ can span one or more words as in the examples above, which effectively means masking one or more words from the original sentence. We limit the length of $sp$ to the maximum of $max_{sp}$ consecutive words.

**Positive and Negative Pairs for CL.** The main idea behind CL in Stage 1 is to adapt the input (multilingual) sentence encoder to the span classification task by 'teaching' it to encode sentences carrying the same slot types closer in its CL-refined semantic space. The pair $p$=($s_{mask}$, $sp$) is extracted from the corresponding tuple, and the encoding of the pair is a concatenation of encodings of $s_{mask}$ and $sp$ encoded separately by the sentence encoder. CL proceeds in a standard fashion relying on sets of positive and negative CL pairs. A positive pair (actually, 'a pair of pairs') is one where two pairs $p_i$ and $p_j$ contain the same label $L$ in their corresponding tuple, but only if $L \neq$ None.[2] A negative pair is one where two pairs $p_i$ and $p_j$ contain different labels $L_i$ and $L_j$ in their tuples, but at least one of the labels is not None.

Following prior CL work (Vulić et al., 2021), each positive pair $(p_i, p_j)$ is associated with $2K$ negative pairs, where we randomly sample $K$ negatives associated with $p_i$ and $K$ negatives for $p_j$. Finally, for the special and most efficient CL setup where the ratio of positive and negative pairs is $1:1$, we first randomly sample the item from the positive pair $(p_i, p_j)$, and then randomly sample a single negative for the sampled $p_i$ or $p_j$.

**Online Contrastive Loss.** Fine-tuning the input sentence encoder with the positive and negative pairs proceeds via a standard *online contrastive loss*. More formally:

$$\mathcal{L}_{contr}(s_i, s_j, f) = \mathbb{1}[y_i = y_j]\|f(s_i) - f(s_j)\|^2 + $$
$$+ \mathbb{1}[y_i \neq y_j]\dot{m}ax(0, m - \|f(s_i) - f(s_j)\|^2)$$

where $s_i$ and $s_j$ are two examples with labels $y_i$ and $y_j$, $f$ is the encoding function and $m$ is a hyperparameter defining the margin between samples of different classes.

Similarly to the original contrastive loss (Chopra et al., 2005), it aims at **1)** reducing the semantic

distance, formulated as the cosine distance, between representations of examples forming the positive pairs, and **2)** increase the distance between representations of examples forming the negative pairs. The online version of the loss, which typically outperforms its standard variant (Reimers and Gurevych, 2019), focuses only on hard positive and hard negative examples: the distance is higher than the margin $m$ for positive examples, and below $m$ for negative examples.

## 3.2 Stage 2: Span Identification and Classification

The aim of Stage 2 is to identify and label the slot spans, relying on the embeddings produced by the encoders fine-tuned in the preceding Stage 1. In order to identify the slot spans, we must consider every possible subspan of the input sentence, which might slow down inference. Therefore, to boost inference speed, we divide Stage 2 into two steps. In Step 1, we perform a simple binary classification, aiming to detected whether a certain span is a slot value for *any* slot type from $\mathcal{S}$. Effectively, for the input pair ($s_{mask}$, $sp$) the binary classifier returns 1 (i.e., '$sp$ is some slot value') or 0. The 0-examples for training are all subspans of the sentences which are not associated with any slot type from $\mathcal{S}$.

Step 2 is a multi-class span classification task, where we aim to predict the actual slot type from $\mathcal{S}$ for the input pair ($s_{mask}$, $sp$). The binary filtering Step 1 allows us to remove all input pairs for which the Step 1 prediction is 0, and we thus assign slot types only for the 1-predictions from Step 1. Put simply, Step 1 predicts if span covers any proper slot value, while Step 2 maps the slot value to the actual slot type. We can directly proceed with Step 2 without Step 1, but the training data then also has to contain all the examples with spans where $L$=None, see Figure 1 again.

The classifiers in both steps are implemented as simple multi-layer perceptrons (MLP), and the input representation in both steps is the concatenation of the respective encodings for $s_{mask}$ and $sp$.

## 4 Experimental Setup

**Training Setup and Data.** The standard few-shot setup in multilingual contexts (Razumovskaia et al., 2022a; Xu et al., 2020) assumes availability of a large annotated task-specific dataset in English, and a handful of labeled examples in the target language. However, as discussed in §1, this as-

---

[2]Put simply, we disallow pulling closer pairs which do not convey any useful slot-related semantic information.

| Dataset | Domains | Slots | Languages | Examples per Lang |
|---------|---------|-------|-----------|-------------------|
| MultiATIS++ | 1 | 84 | de, fr, pt, tr, hi | 5,871 |
| xSID | 7 | 33 | ar, da, de, de-st, id, it, ja, kk, nl, sr, tr | 800 |

Table 1: Multilingual SL datasets in the experiments.

sumption might not always hold. That is, the English data might not be available for many target-language specific domains, especially since the annotation for the SL task is also considered more complex than for the intent detection task (van der Goot et al., 2021; Xu et al., 2020; FitzGerald et al., 2022). We thus focus on training and evaluation in these challenging transfer-free setups.

We run experiments on two standard multilingual SL datasets, simulating the transfer-free setups: MultiATIS++ (Xu et al., 2020) and xSID (van der Goot et al., 2021). Their data statistics are provided in Table 1, with language codes in Appendix A. For low-resource scenarios, we randomly sample $M$ annotated sentences from the full training data. Since xSID was originally intended only for testing zero-shot cross-lingual transfer, we use its limited dev set for (sampling) training instances.

A current limitation of TWOSL is that it leans on whitespace-based word token boundaries in the sentences: therefore, in this work we focus on a subset of languages with that property, leaving further adaptation to other languages for future work.

**Input Sentence Encoders.** We experiment both with multilingual sentence encoders as well as general multilingual PLMs in order to (i) demonstrate the effectiveness of TWOSL irrespective of the underlying encoder, and to (ii) study the effect of pretraining task on the final performance. **1)** *XLM-R* (Conneau et al., 2020) is a multilingual PLM, pretrained with a large multilingual dataset in 100 languages via masked language modeling. **2)** Multilingual *mpnet* (Song et al., 2020) is pretrained for paraphrase identification in over 50 languages; the model was specifically pretrained in a contrastive fashion to effectively encode sentences. **3)** We also run a subset of experiments with another state-of-the-art multilingual sentence encoder, *LaBSE* (Feng et al., 2022), to further verify that TWOSL can be disentangled from the actual encoder.[3] All

models are used in their 'base' variants, with 12 hidden-layers and encoding the sequences into 768-dimensional vectors. This means that the actual encodings of $(s_{mask}, sp)$ pairs, which are fed to MLPs in Stage 2, are 1,536-dimensional; see §3.

**Hyperparameters and Optimisation.** We rely on *sentence-transformers* (SBERT) library (Reimers and Gurevych, 2019, 2020) for model checkpoints and contrastive learning in Stage 1. The models are fine-tuned for 10 epochs with batch size of 32 using the default hyperparameters in SBERT: e.g., the margin in the contrastive loss is fixed to $m = 0.5$. $max_{sp}$ is fixed to 5 as even the longest slot values very rarely exceed that span length. Unless stated otherwise, $K = 1$, that is, the ratio of positive-to-negative examples is $1 : 2$, see §3.1.

In Stage 2, we train binary and multi-class MLPs with the following number of hidden layers and their size, respectively: [2,500, 1,500] and [3,600, 2,400, 800], and ReLU as the non-linear activation. The Step 1 binary classifier is trained for 30 epochs, while the Step 2 MLP is trained for 100 epochs. The goal in Step 1 is to ensure high recall (i.e., to avoid too aggressive filtering), which is why we opt for the earlier stopping. As a baseline, we fine-tune XLM-R for the token classification task, as the standard SL task format (Xu et al., 2020; Razumovskaia et al., 2022b). Detailed training hyperparameters are provided in Appendix B. All results are averages across 5 random seeds.

**Evaluation Metric.** For direct comparability with standard token classification approaches we rely on token-level micro-$F_1$ as the evaluation metric. For TWOSL this necessitates the reconstruction of the BIO-labeled sequence $Y$ from the predictions for the $(s_{mask}, sp, L_{pred})$ tuples. For every sentence $s$ we first identify all the tuples $(s_{mask}, sp, L_{pred})$ associated with $s$ such that the predicted slot type $L_{pred} \neq$ None. In $Y$ the positions of $sp$ are filled with $B_{L_{pred}}$, complemented with the corresponding number of $I_{L_{pred}}$ if the length of $sp > 1$. Following that, the rest of the positions are set to the $O$ label.

## 5 Results and Discussion

Before experimenting in the planned multilingual context, we evaluate our model in English, based on the standard ATIS dataset. The models are trained with the hyper-parameters described in §4. For English, we use LaBSE (Feng et al., 2022) as a sequence encoder as it has demonstrated state-of-the-art results in prior experiments in dialogue-specific

---

[3]Prior work has demonstrated effectiveness of models pretrained for span encoding in few-shot settings (Henderson and Vulić, 2021; Coope et al., 2020). However, they are not directly comparable with TWOSL as they are based on English-only encoders and are not publicly available.

|  | AR | DE | DA | DE-ST | ID | IT | KK | NL | SR | TR | AVG |
|---|---|---|---|---|---|---|---|---|---|---|---|
| **50 training examples** | | | | | | | | | | | |
| XLM-R | 3.1 | 2.2 | 5.0 | 0.7 | 10.4 | 1.3 | 12.5 | 0.7 | 0.7 | 5.5 | 4.2 |
| XLM-R-Sent w/o CL | 12.7 | 23.9 | 22.5 | 27.2 | 23.3 | 26.9 | 20.1 | 29.5 | 22.7 | 25.7 | 23.5 |
| XLM-R-Sent w/ CL | **40.2** | **49.5** | 41.2 | 45.2 | **46.7** | **49.2** | 43.1 | 36.1 | 41.5 | 44.7 | **43.7** |
| mpnet w/o CL | 23.9 | 25.1 | 26.8 | 21.8 | 28.5 | 26.3 | 22.6 | 29.7 | 23.9 | 24.5 | 25.3 |
| mpnet w/ CL | 36.8 | 40.4 | 39.2 | 39.1 | 43.4 | 42.2 | 33.5 | **38.8** | 39.7 | 42.8 | 39.6 |
| **100 training examples** | | | | | | | | | | | |
| XLM-R | 33.0 | 37.1 | 36.0 | 32.5 | 39.6 | 37.4 | 31.1 | 37.7 | 34.4 | 37.7 | 35.6 |
| XLM-R-Sent w/o CL | 28.1 | 33.2 | 30.9 | 29.2 | 30.6 | 36.3 | 29.4 | 41.5 | 27.0 | 34.2 | 32.0 |
| XLM-R-Sent w/ CL | 39.0 | 46.2 | 42.9 | **45.0** | 51.4 | 36.4 | **41.6** | 52.1 | 35.7 | 50.4 | 44.1 |
| mpnet w/o CL | 34.3 | 37.1 | 36.3 | 31.2 | 37.6 | 38.6 | 30.0 | 37.1 | 33.7 | 32.8 | 34.9 |
| mpnet w/ CL | **43.5** | **51.1** | 44.2 | 44.1 | **52.6** | **47.2** | 40.4 | 48.3 | **46.5** | **51.1** | **46.9** |
| **200 training examples** | | | | | | | | | | | |
| XLM-R | 39.5 | 45.2 | 43.1 | 41.3 | 47.8 | 44.8 | 37.5 | 44.0 | 42.0 | 45.1 | 43.0 |
| mpnet w/o CL | 41.3 | 44.9 | 44.9 | 42.2 | 46.5 | 46.5 | 37.8 | 46.3 | 41.7 | 41.2 | 43.3 |
| mpnet w/ CL | **48.2** | **55.8** | **50.3** | **52.1** | **59.0** | **55.2** | **46.1** | **53.8** | **52.9** | **52.5** | **52.6** |

Table 2: Results on xSID's test set using a subsample of 50, 100, and 200 examples from its validation portion for training, with no English training examples. Micro $F_1$ scores are reported. *XLM-R* refers to using the XLM-R PLM for the standard token classification fine-tuning for SL. *XLM-R-Sent* denotes using XLM-R directly as a sentence encoder in the same fashion as *mpnet*. We provide standard deviation for results with *mpnet* as the sentence encoder in Appendix D, demonstrating statistical significance of the improvements provided by contrastive learning.

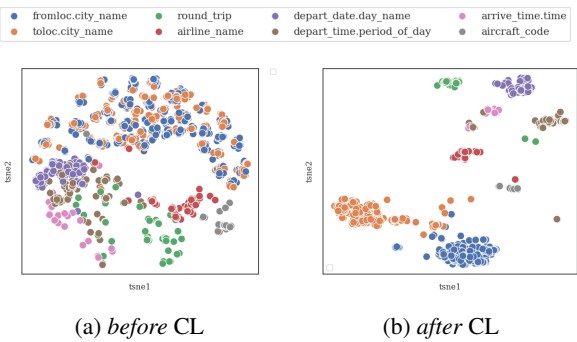

(a) *before* CL       (b) *after* CL

Figure 2: t-SNE plots (van der Maaten and Hinton, 2012) for annotated German examples from Multi-ATIS++'s test set. We show the examples for 8 slot types, demonstrating the effect of Contrastive Learning (CL) on the final encodings. The encodings were created using **(a)** the original mpnet encoder before Stage 1 CL and **(b)** mpnet after CL-tuning in Stage 1. 800 annotated training examples were used for CL, $K = 1$.

tasks (Casanueva et al., 2022). The results in Table 3 demonstrate that TWOSL is effective in low-data English-only setups, with large gains even atop such a strong sentence encoder as LaBSE. This indicates that TWOSL can be used to bootstrap any project when only a handful of in-domain data points are available.

**Impact of Contrastive Learning in TWOSL on Slot Representations.** Further, before delving deep into quantitative analyses, we investigate what

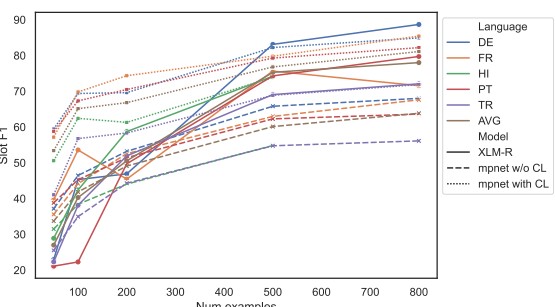

Figure 3: Slot $F_1$ scores on MultiATIS++ across different languages and data setups. The exact numerical scores are available in Appendix C.

|  | 50 | 100 | 200 |
|---|---|---|---|
| XLM-R | 28.60 | 47.14 | 73.48 |
| TWOSL: LaBSE w/o CL | 41.88 | 52.14 | 65.01 |
| TWOSL: LaBSE w/ CL | 64.92 | 76.17 | 82.51 |

Table 3: Results on English ATIS in few-shot setups with 50, 100, and 200 training examples.

effect CL in Stage 1 of TWOSL actually has on span encodings, and how it groups them over slot types. The aim of CL in Stage 1 is exactly to make the representations associated with particular cluster into coherent groups and to offer a clearer separation of encodings across slot types. As proven previously for the intent detection task (Vulić et al., 2021), such well-divided groups in the encoding space might facilitate learning classifiers on top of

the fine-tuned encoders. As revealed by a t-SNE plot (van der Maaten and Hinton, 2012) in Figure 2, which shows the *mpnet*-based encodings before and after Stage 1, exactly that effect is observed.

Namely, the non-tuned mpnet encoder already provides some separation of encodings into slot type-based clusters (Figure 2a), but the groupings are less clear and noisier. In contrast, in Figure 2b the examples are clustered tightly by slot type, with clear separations between different slot type-based clusters. This phenomenon is further corroborated by the automated Silhouettes cluster coherence metric (Rousseeuw, 1987): its values are $\sigma = -0.02$ (before Stage 1) and $\sigma = 0.67$ (after Stage 1). In sum, this qualitative analysis already suggests the effectiveness of CL for the creation of customised span classification-oriented encodings that support the SL task. We note that the same observations hold for all other languages as well as for all other (data-leaner) training setups.

**Main Results.** The results on xSID and Multi-ATIS++ are summarised in Table 2 and Figure 3, respectively. The scores underline three important trends. First, TWOSL is much more powerful than the standard PLM-based (i.e., XLM-R-based) token classification approach in very low-data setups, when only a handful (e.g., 50-200) annotated examples in the target language are available as the only supervision. Second, running TWOSL on top of a general-purpose multilingual encoder such as mpnet yields large and consistent gains, and this is clearly visible across different target languages in both datasets, and across different data setups. Third, while the token classification approach is able to recover some performance gap as more annotated data become available (e.g., check Figure 3 with 800 examples), TWOSL remains the peak-performing approach in general.

A finer-grained inspection of the scores further reveals that for low-data setups, even when exactly the same model is used as the underlying encoder (i.e., XLM-R), TWOSL offers large benefits over token classification with full XLM-R fine-tuning, see Table 2. The scores also suggest that the gap between TWOSL and the baselines increases with the decrease of annotated data. The largest absolute and relative gains are in the 50-example setup, followed by the 100-example setup, etc.: e.g., on xSID, the average gain is +9.5 F1 points with 200 training examples, while reaching up to +35.3 F1 points with 50 examples. This finding corroborates

|  | 50 | | 100 | | 200 | | 500 | |
|---|---|---|---|---|---|---|---|---|
|  | DE | FR | DE | FR | DE | FR | DE | FR |
| XLM-R | **82.4** | 74.5 | **83.4** | **75.5** | **87.9** | 78.3 | **89.8** | **85.1** |
| TWOSL: mpnet w/o CL | 56.3 | 55.2 | 62.9 | 57.7 | 68.5 | 63.8 | 71.2 | 70.2 |
| TWOSL: mpnet w/ CL | 82.1 | **75.2** | 76.8 | 71.8 | 84.7 | **78.9** | 87.6 | 84.6 |

Table 4: Results on German and French in MultiATIS++ for standard few-shot setup where English annotated data is combined with a few target language examples.

the power of CL especially for such low-data setups. Finally, the results in Table 2 also hint that TWOSL works well with different encoders: it improves both mpnet and XLM-R as the underlying multilingual sentence encoders.

### 5.1 Ablations and Further Analyses

**TWOSL for Low-Resource Languages.** TWOSL is proposed for extremely low-resource scenarios, when only a handful of examples in a target language are available. This is more likely to happen for low-resource languages, which in addition are usually not represented enough in pre-training of PLMs (Conneau et al., 2020). To evaluate TWOSL on low-resource languages, we apply it to Named Entity Recognition (NER) task in several low-resource African languages. We focused on NER as i) similarly to slot labelling, NER is a sequence labelling task; ii) limited annotated data is available in low-resource languages for NER. Specifically, we use MasakhaNER (Adelani et al., 2021) dataset in our experiments focusing on Yoruba (*yor*) and Luo (*luo*). The experiments were conducted with 100 training examples, using XLM-R (base) as a sentence encoder. The rest of the setup was kept the same as described in §4.

TWOSL has brought considerable improvements for both languages: from 14.46 to 45.38 F-1 for *yor* and from 16.56 to 36.46 F-1 for *luo*. These results further indicate the effectiveness of the method for low-resource languages as well as for language-domain combinations with scarce resources.

**TWOSL in Standard Few-Shot Setups.** TWOSL has been designed with a primary focus on transfer-free, extremely low-data setups. However, another natural question also concerns its applicability and effectiveness in the standard few-shot transfer setups, where we assume that a large annotated dataset for the same task and domain is available in the source language: English. To this end, we run several experiments on MultiATIS++, with German and French as target languages, where we first fine-

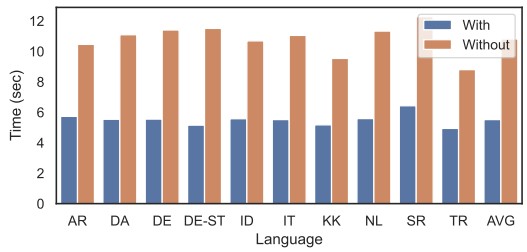

Figure 4: Inference time per language on XSID with and without the binary filtering Step 1 in Stage 2.

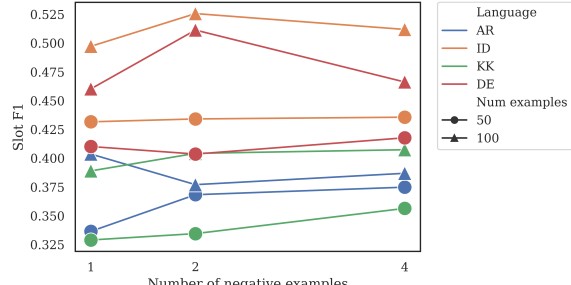

Figure 5: Impact of the number of negative examples per each positive example for the 50-example and 100-example setups. For clarity, the results are shown for a subset of languages in xSID, and the similar trends are observed for other languages. Similar trends are observed on MultiATIS++, as shown in App. F.

| | AR | DA | ID | IT | KK | SR | TR | AVG |
|---|---|---|---|---|---|---|---|---|
| **50 examples** | | | | | | | | |
| XLM-R-Sent w/o CL | 12.7 | 22.5 | 23.3 | 26.9 | 20.1 | 22.7 | 25.7 | 22.0 |
| XLM-R-Sent w/ CL | 40.2 | 41.2 | 46.7 | **49.2** | **43.1** | **41.5** | **44.7** | **43.8** |
| mpnet w/o CL | 23.9 | 26.8 | 28.5 | 26.3 | 22.6 | 23.9 | 24.8 | 25.2 |
| mpnet w/ CL | 36.8 | 39.2 | 43.4 | 42.2 | 33.5 | 39.7 | 42.8 | 39.6 |
| LaBSE w/o CL | 21.4 | 31.7 | 33.7 | 31.6 | 29.0 | 25.5 | 27.7 | 28.6 |
| LaBSE w/ CL | **41.8** | **47.0** | **48.2** | 48.3 | 41.8 | 36.6 | 37.9 | 43.1 |
| **100 examples** | | | | | | | | |
| XLM-R-Sent w/o CL | 28.1 | 30.9 | 30.6 | 36.3 | 29.4 | 27.0 | 34.2 | 30.9 |
| XLM-R-Sent w/ CL | 39.0 | 42.9 | 51.4 | 36.4 | 41.6 | 35.7 | 50.4 | 42.5 |
| mpnet w/o CL | 34.3 | 36.3 | 37.6 | 38.7 | 30.0 | 33.7 | 32.8 | 34.8 |
| mpnet w/ CL | 43.5 | 44.2 | **52.6** | 47.2 | 40.4 | **46.5** | **51.1** | 46.5 |
| LaBSE w/o CL | 31.7 | 40.7 | 37.9 | 38.5 | 34.9 | 37.1 | 37.1 | 36.8 |
| LaBSE w/ CL | **47.3** | **50.2** | 49.6 | **53.6** | **45.5** | 39.4 | 42.8 | **46.9** |

Table 5: Results on xSID for a sample of languages with different sentence encoders. XLM-R-Sent denotes using XLM-R as a standard sentence encoder.

tune the model on the full English training data, before running another fine-tuning step (Lauscher et al., 2020) on the $M = 50, 100, 200, 500$ examples in the target language.

Overall, the results in Table 4 demonstrate that TWOSL maintains its competitive performance, although the token classification approach with XLM-R is a stronger method overall in this setup. TWOSL is more competitive for French as the target language. The importance of CL in Step 1 for TWOSL is pronounced also in this more abundant data setup. We leave further exploration and adaptation of TWOSL to transfer setups for future work.

**Impact of Binary Filtering in Stage 2.** In order to understand the benefit of Step 1 (i.e., binary filtering) in Stage 2, we compare the performance and inference time with and without that step. We focus on the xSID dataset in the 200-example setup. The scores, summarised in Table 10 in Appendix E, demonstrate largely on-par performance between the two variants. The main benefit of using Step 1 is thus its decrease of inference time, as reported in Figure 4, where inference was carried out on a single NVIDIA Titan XP 12GB GPU. The filtering step, which relies on a more compact and thus quicker classifier, greatly reduces the number of examples that have to undergo the final, more expensive slot type prediction (i.e., without filtering all the subspans of the user utterance must be processed) without harming the final performance.

**Different Multilingual Encoders.** The results in Table 2 have already validated that TWOSL offers gains regardless of the chosen multilingual encoder (e.g., XLM-R versus mpnet). However, the effectiveness of TWOSL in terms of absolute scores is naturally dependent on the underlying multilingual capabilities of the original multilingual encoder. We thus further analyse how the performance changes in the same setups with different

encoders. We compare XLM-R-Sent (i.e., XLM-R used a sentence encoder, mean-pooling all subword embeddings), mpnet, and LaBSE on a representative set of 7 target languages on xSID. In the majority of the experimental runs, LaBSE with TWOSL yields the highest absolute scores. This comes as no surprise as LaBSE was specifically customised to improve sentence encodings for low-resource languages and in low-resource setups (Feng et al., 2022). Interestingly, XLM-R performs the best in the 'lowest-data' 50-example setup: we speculate this might be due to a smaller model size, which makes it harder to overfit in extremely low-resource setups. Finally, the scores again verify the benefit of TWOSL when applied to any underlying encoder.

**Number of Negative Pairs.** The ratio of positive-to-negative examples, controlled by the hyperparameter $K$, has a small impact on the overall performance, as shown in Figure 5. We observe some slight performance gains when moving from 1 neg-

ative example to 2 (cf., the 50-example setup for AR and 100-example setup for ID in xSID). In such cases, the increase in the number of negative pairs can act as data augmentation for the extreme low-resource scenarios. This hyper-parameter also impacts the trade-off between training time and the stability of results. With fewer negative examples, training is quicker, but the performance is less stable: e.g., in the 50-example setup for German in MultiATIS++, the standard deviation is $\sigma = 7.45$, $\sigma = 2.36$ and $\sigma = 3.21$ with 1,2 and 4 negatives per positive, respectively. Therefore, as stated in §4, we use the setup with 2 negatives-per-positive in our experiments, indicating the good trade-off between efficiency and stability.

## 6 Conclusion and Future Work

We proposed TWOSL, a two-stage slot labeling approach which turns multilingual sentence encoders into slot labelers for task-oriened dialogue (ToD), which was proven especially effective for slot labeling in low-resource setups and languages. TWOSL was developed with the focus on *transfer-free* few-shot multilingual setups, where sufficient English-language annotated data are not readily available to enable standard cross-lingual transfer approaches. In other words, the method has been created for bootstrapping a slot labeling system in a new language and/or domain when only a small set of annotated examples is available. TWOSL first converts multilingual sentence encoders into task-specific span encoders via contrastive learning. It then casts slot labeling into the span classification task supported by the fine-tuned encoders from the previous stage. The method was evaluated on two standard multilingual ToD datasets, where we validated its strong performance across diverse languages and different training data setups.

Due to its multi-component nature, a spectrum of extensions focused on its constituent components is possible in future work, which includes other formulations of contrastive learning, tuning the models multilingually, mining (non-random) negative pairs and extending the method to cross-domain transfer learning for ToD (Majewska et al., 2022), especially for rare domains not covered by standard datasets. In this work, we have focused on the sample-efficient nature of TWOSL. We expect it to be complementary to modular and parameter-efficient techniques (Pfeiffer et al., 2023). In the long run, we plan to use the method for large-scale fine-tuning of sentence encoders to turn them into universal span encoders which can then be used on sequence labelling tasks across languages and domains. TWOSL can be further extended to slot labelling with nested slots as well as to other 'non-ToD' sequence labelling tasks (e.g., NER) for which evaluation data exists for truly low-resource languages: e.g., on African languages (Adelani et al., 2021).

## Limitations

TWOSL relies on whitespace-based word boundaries. Thus, it is only applicable to languages which use spaces as word boundaries. We plan to extend and adapt the method to other languages, without this property, in our subsequent work. Additionally, the approach has been only tested on the languages which the large multilingual PLMs have seen during their pretraining. We plan to adapt and test the same approach on unseen languages in the future.

As mentioned in §6, we opted for representative multilingual sentence encoders and components of contrastive learning that were proven to work well for other tasks in prior work (Reimers and Gurevych, 2020; Vulić et al., 2021) (e.g., the choice of the contrastive loss, adopted hyper-parameters), while a wider exploration of different setups and regimes in TWOSL's Stage 1 and Stage 2 might further improve performance and offer additional low-level insights.

The scope of our multilingual evaluation is also constrained by the current availability of multilingual evaluation resources for ToD NLU tasks.

Finally, in order to unify the experimental protocol across different languages, and for a more comprehensive coverage and cross-language comparability, we relied on multilingual encoders throughout the work. However, we stress that for the transfer-free scenarios, TWOSL is equally applicable to monolingual encoders for respective target languages, when such models exist, and this might yield increased absolute performance.

## Acknowledgments

The work was in part supported by a Huawei research donation to the Language Technology Lab at the University of Cambridge. Ivan Vulić is also supported by a personal Royal Society University Research Fellowship *'Inclusive and Sustainable Language Technology for a Truly Multilingual World'* (no 221137; 2022–).

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

# A    Language codes

Language codes which are used in the paper are provided in Table 6.

| Language code | Language |
|---|---|
| ar | Arabic |
| da | Danish |
| de | German |
| de-st | German South-Tyrolean |
| fr | French |
| hi | Hindi |
| id | Indonesian |
| it | Italian |
| ja | Japanese |
| kk | Kazakh |
| nl | Dutch |
| pt | Portuguese |
| sr | Serbian |
| tr | Turkish |

Table 6: Language codes used in the paper

# B    Training Hyperparameters

The values for training hyperparameters for each stage of TWOSL and token classification are provided in Table 7. For information about Adam and AdamW optimizers we refer the reader to Kingma and Ba (2015) and Loshchilov and Hutter (2019), respectively.

| Model | Epochs | Batch | Optim | LR | WD |
|---|---|---|---|---|---|
| Token classification | 50 | 32 | Adam | $1e-5$ | 0.01 |
| TWOSL: Stage 1 | 10 | 32 | AdamW | $2e-5$ | 0.01 |
| TWOSL: Stage 2, Step 1 | 30 | 32 | Adam | $1e-5$ | 0.0 |
| TWOSL: Stage 2, Step 2 | 100 | 32 | Adam | $1e-5$ | 0.0 |

Table 7: Training hyperparameters for token classification and TWOSL. Acronyms: LR – learning rate, WD – weight decay rate.

# C    Full Scores on MultiATIS++

The exact numerical scores on MultiATIS++ across different languages and setups, which were used as the source for Figure 3 in the main paper, are provided in Table 8.

# D    Standard Deviation for xSID results using mpnet as a sentence encoder

The standard deviation of 5 runs for xSID are presented in Table 9. The standard deviation which is considerably lower than the margin between the performance of systems without and with contrastive learning, proving the significance of improvements that CL provides.

|              | DE   | FR   | HI   | PT   | TR   | AVG  |
|--------------|------|------|------|------|------|------|
| **50 examples** | | | | | | |
| XLM-R        | 22.9 | 39.6 | 28.9 | 21.1 | 22.3 | 27.0 |
| mpnet w/o CL | 37.2 | 35.6 | 31.5 | 38.8 | 25.5 | 33.7 |
| mpnet w/ CL  | **59.4** | **57.2** | **50.6** | **58.8** | **41.1** | **53.4** |
| **100 examples** | | | | | | |
| XLM-R        | 45.3 | 53.6 | 42.4 | 22.3 | 38.1 | 40.3 |
| mpnet w/o CL | 46.5 | 44.2 | 38.4 | 45.3 | 35.0 | 41.9 |
| mpnet w/ CL  | **69.4** | **69.8** | **62.4** | **67.3** | **56.8** | **65.1** |
| **200 examples** | | | | | | |
| XLM-R        | 46.9 | 45.4 | 58.7 | 49.7 | 51.9 | 50.6 |
| mpnet w/o CL | 53.2 | 52.6 | 44.0 | 51.4 | 44.3 | 49.1 |
| mpnet w/ CL  | **69.6** | **74.4** | **61.3** | **70.5** | **58.3** | **66.8** |
| **500 examples** | | | | | | |
| XLM-R        | **83.1** | 75.6 | **74.2** | 74.3 | **69.1** | 75.3 |
| mpnet w/o CL | 65.8 | 63.0 | 54.8 | 62.3 | 54.8 | 60.1 |
| mpnet w/ CL  | 82.2 | **79.8** | 73.8 | **79.3** | 68.8 | **76.8** |
| **800 examples** | | | | | | |
| XLM-R        | **88.7** | 71.6 | N/A | 79.7 | **72.1** | 78.0 |
| mpnet w/o CL | 68.0 | 67.6 | N/A | 63.6 | 56.1 | 63.8 |
| mpnet w/ CL  | 84.9 | **85.4** | N/A | **82.2** | 71.9 | **81.1** |

Table 8: Results on MultiATIS++. $K = 1$. The results are averaged across 5 random seeds.

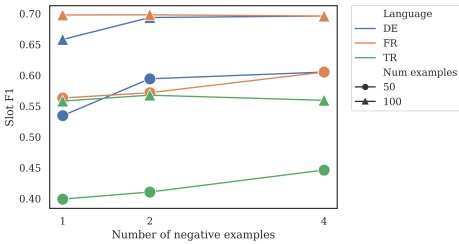

Figure 6: Impact of the number of negative examples per each positive example for the 50-example and 100-example setups. For clarity, the results are shown for a subset of languages in MultiATIS++, and the similar trends are observed for other languages.

# E   Results on xSID with and without the Binary Filtering Step

A comparison of results with and without applying the binary filtering step (i.e., Step 1 in Stage 2 of TWOSL) is provided in Table 10; see §5.1 for the discussion supported by this set of results.

# F   Number of negative examples for MultiATIS++

| | AR | DE | DA | DE-ST | ID | IT | KK | NL | SR | TR | AVG |
|---|---|---|---|---|---|---|---|---|---|---|---|
| **50 training examples** | | | | | | | | | | | |
| mpnet w/o CL | 1.88 | 3.14 | 1.79 | 2.91 | 2.55 | 1.52 | 2.51 | 1.82 | 3.87 | 2.42 | 2.44 |
| mpnet w/ CL | 3.04 | 1.70 | 4.18 | 2.53 | 11.72 | 2.97 | 2.86 | 1.93 | 1.95 | 3.14 | 3.60 |
| **100 training examples** | | | | | | | | | | | |
| mpnet w/o CL | 2.74 | 2.01 | 1.79 | 3.65 | 2.29 | 1.11 | 2.29 | 3.48 | 1.07 | 0.81 | 2.12 |
| mpnet w/ CL | 3.18 | 2.96 | 2.64 | 3.25 | 1.71 | 1.89 | 1.62 | 0.95 | 5.62 | 2.69 | 2.65 |
| **200 training examples** | | | | | | | | | | | |
| mpnet w/o CL | 1.72 | 1.28 | 1.94 | 1.84 | 2.41 | 0.96 | 1.41 | 2.40 | 4.17 | 1.74 | 1.99 |
| mpnet w/ CL | 3.99 | 4.87 | 1.72 | 2.01 | 2.44 | 6.58 | 1.75 | 3.52 | 3.18 | 0.63 | 3.06 |

Table 9: Standard deviation on xSID's test set using a subsample of 50, 100, and 200 examples from its validation portion for training. The standard deviation is calculated for 5 random seed settings.

| | AR | DE | DA | DE-ST | ID | IT | KK | NL | SR | TR | AVG |
|---|---|---|---|---|---|---|---|---|---|---|---|
| **50 examples** | | | | | | | | | | | |
| mpnet w/o CL w/ step 1 | 23.9 | 25.1 | 26.8 | **21.8** | **28.5** | 26.3 | 22.6 | **29.7** | 23.9 | 24.8 | **25.3** |
| mpnet w/o CL w/o step 1 | **24.8** | 25.5 | 28.3 | 21.6 | 28.4 | **26.7** | 23.1 | 26.9 | 22.8 | 24.0 | 25.2 |
| mpnet w/ CL w/ step 1 | 36.8 | 40.4 | 39.2 | 39.1 | 43.4 | 42.2 | 33.5 | 38.8 | 39.7 | **42.8** | 39.6 |
| mpnet w/ CL w/o step 1 | **37.0** | **41.6** | **41.7** | **40.4** | **44.8** | 40.8 | 35.2 | 41.4 | 39.7 | 40.8 | **40.3** |
| **100 examples** | | | | | | | | | | | |
| mpnet w/o CL w/ step 1 | 34.3 | **37.1** | **36.2** | 31.2 | 37.6 | 38.7 | 30.0 | 37.1 | 33.7 | 32.8 | **34.9** |
| mpnet w/o CL w/o step 1 | **34.5** | 35.6 | 34.6 | 30.5 | **37.5** | 35.6 | 28.1 | 36.5 | 33.5 | **33.7** | 34.0 |
| mpnet w/ CL w/ step 1 | **43.5** | **51.1** | 44.2 | 44.1 | **52.6** | 47.2 | 40.4 | **48.3** | **46.5** | **51.1** | **46.9** |
| mpnet w/ CL w/o step 1 | 42.7 | 50.3 | **44.9** | **46.2** | 48.6 | **48.0** | **41.0** | 47.5 | 45.3 | 47.6 | 46.2 |
| **200 examples** | | | | | | | | | | | |
| mpnet w/o CL w/ step 1 | 41.3 | 44.9 | **44.9** | 42.2 | 46.5 | **46.5** | 37.8 | 46.3 | 41.7 | **41.2** | 43.3 |
| mpnet w/o CL w/o step 1 | **42.8** | **45.8** | 44.4 | **43.3** | **47.6** | 46.2 | 36.6 | **46.9** | **43.4** | 41.3 | **43.8** |
| mpnet w/ CL w/ step 1 | **48.2** | **55.8** | 50.3 | **52.1** | 59.0 | **55.2** | **46.1** | 53.8 | 52.9 | **52.5** | **52.6** |
| mpnet w/ CL w/o step 1 | 47.3 | 52.3 | **51.6** | 51.9 | **60.8** | 54.0 | 45.3 | **54.5** | **53.4** | 52.6 | 52.4 |

Table 10: Results on xSID across different languages and setups with and without applying the binary filtering step: Step 1 in Stage 2 of TWOSL.