# OpenReview forum: "Transfer-Free Data-Efficient Multilingual Slot Labeling"
_EMNLP/2023/Conference — EMNLP 2023 Main_

### Official Review · Reviewer_N33Q · 2023-07-26

**Soundness:** 4

**Excitement:**

4: Strong: This paper deepens the understanding of some phenomenon or lowers the barriers to an existing research direction.

**Missing References:**

For the related work section, there are also slot projection methods that do not require word alignment and rely on a small amount of data and multilingual PLMs [1].

[1] Translate & Fill: Improving Zero-Shot Multilingual Semantic Parsing with Synthetic Data


**Paper Topic And Main Contributions:**

The paper proposes a parsing approach that works better than standard token classification in scarce data multilingual scenarios. Instead of using the few examples available for each language to train a token classification model, i.e. outputting a slot label for each input token, they decompose the parsing task into two classification tasks. The authors first train a binary classifier to identify which among the spans that can be extracted from a sentence may be a slot value. This classifier is used to filter the n^2 spans generated from a sentence of length n and reduce the inference time of the second classification step. For the latter, the authors train a sentence encoder using a contrastive objective that pushes the representations of slot values having the same (or different) label closer (or farther). The sentence encoder updated with this contrastive loss is further fine-tuned on slot classification data and used to label the spans from the sentence.

This method works better than token classification especially when there is little annotated data for each language. The authors convincingly show that the contrastive loss step is crucial to obtain better performance. The quality of the base sentence model affects the final performance motivating for increasingly better base models.

The paper is missing a seq2seq baseline powered by some pretrained multilingual LM (mT5 or mBart, for example), which have become standard for parsing tasks.

**Questions For The Authors:**

In general, zero-shot models trained on a single language are penalised by annotation guidelines that may be inconsistent across languages, for example including or excluding determiners or prepositions in a slot value. Having a small number of examples for each language helps in fixing language specific annotation inconsistencies. A model trained on a single language will default to the same behavior for other languages, most of the time.

In the Table 2 experiment, it is not clear if and how English data is used to train the model.

Figure 4: is the “with” column the sum of the binary and slot classifier times, or just the inference time of the latter on the XSID examples?

What about seq2seq methods for parsing such as mT5 or ByT5?


**Reasons To Accept:**

The paper proposes a sample-efficient method for parsing that requires little annotated data to perform reasonably well in a multilingual setting.

Research on few-shot and scarce data scenarios, especially considering low-resource languages, is extremely important.


**Reasons To Reject:**

There is a risk that results from multilingual seq2seq baselines could affect the paper story.


**Reproducibility:**

4: Could mostly reproduce the results, but there may be some variation because of sample variance or minor variations in their interpretation of the protocol or method.

**Reviewer Confidence:**

4: Quite sure. I tried to check the important points carefully. It's unlikely, though conceivable, that I missed something that should affect my ratings.

---

> ### Author Rebuttal · Authors · 2023-08-29
>
> We are extremely grateful for your in-depth review and your thoughts and suggestions for the paper.
>
> With regards to the seq2seq baselines, thanks for the suggestion of an additional baseline! We will add a full set of results to the final version of the paper. However, our preliminary results with mT5 demonstrated that seq2seq formulation with mT5 as a base model is only marginally better than the standard supervised model.
>
> With regards to the questions, we agree that TWOSL improves cross-lingual performance both by avoiding source language inductive bias and by avoiding the discrepancies between languages in annotation guidelines.
>
> In the experiment in Table 2 the training data did not include any English data. The models were trained only using the few-shot target language data. We will make sure to add further clarifications in the corresponding (Main Results) subsection of the paper. Table 4 provides some results in transfer-based scenarios where we assume availability of large annotated data in English and we perform few-shot transfer.
>
> In Fig. 4, the bars “With” included the total time it took to classify the spans, i.e., the sum of binary classifier and slot classifier. The reduction in time processing arises from the difference in the processing of inputs: in the setup “With” binary classifier all sentence-span pairs are processed by the fast binary classifier and full slot classifier processes only a subset of sentence-span pairs which are classified as related by the binary classifier. At the same time, in “Without” setup all the sentence span pairs go through the slot classifier.

---

### Official Review · Reviewer_7N7r · 2023-08-01

**Soundness:** 4

**Excitement:**

4: Strong: This paper deepens the understanding of some phenomenon or lowers the barriers to an existing research direction.

**Missing References:**

Works on Multilingual Slot Labeling / Semantic Parsing which might be of interest:

https://arxiv.org/pdf/2204.08582v2.pdf (Task-Oriented Dialogue Dataset that would allow for extensions to more typologically diverse languages)

https://aclanthology.org/2022.acl-long.285/ and https://aclanthology.org/2023.acl-long.199/ (Adversarial Alignment methods for zero-shot slot labeling/semantic parsing)

https://aclanthology.org/2021.findings-emnlp.279.pdf (State of the art label projection method for translation and slot filling)

Not missing references, but things which would be stronger baselines:

https://arxiv.org/abs/2205.05638 (Existing work for low-data tasks vs. the full finetuning baseline used)

https://aclanthology.org/2021.acl-long.172.pdf (Adapters are more robust to overfitting, whereas full finetuning seems likely to overfit in these low-data scenarios)

**Paper Topic And Main Contributions:**

This work looks at developing an effective method for bootstrapping a slot filling model using limited annotated data across multiple languages. They first refine a pretrained sentence encoder for slot filling using a span-contrastive loss with hard positive and hard negative sampling. Then, they train a two step classifier. The first step classifies whether a span is a valid slot at all, while the second step predicts which slot type the span falls under. They show this procedure is empirically effective for bootstrapping slot filling models using MultiATIS and xSID for XLM-R, mpnet, and LaBSE.

**Questions For The Authors:**

A) Given that the method doesn't rely on transfer, is there a need to use massively Multilingual PLM's for this method? For languages where a monolingual (or more specific language family based PLM such as AfriBerta or Bertic) exists, would they be better choices since they avoid negative transfer from pretraining?

B) Could the method be applied in scenarios where transfer data does exist? It doesn't seem uniquely "transfer-free", so might have broader use as a data efficiency method for transfer scenarios as well right? It would also be interesting to see if this method continues to be advantageous as labeled data continues to scale beyond 200 examples.

C) Is the method applicable to annotation structures which have nested slots? It seems like perhaps no since stage 2 assigns only one slot per token?

D) You mention that as a limitation the method relies on white-space separated words. What about the method prevents you from using the outputs of a tokenizer directly?

**Reasons To Accept:**

- The work poses and tackles an interesting challenge of a learning a slot filling model for a novel annotation scheme. This is well justified since the granularity of slot filling tasks often makes them less transferable than more universal span filling tasks like NER, especially across linguistic and cultural contexts. Overall, it seems like improving capabilities in this regard could enable into tools which enable new applications of NLP.

- The work is quite thorough in showing the effects of their approach across multiple languages, model families, and data scales. The effect is consistently significant across settings. Each component of the approach is shown to be valuable through targeted ablations which motivates the entire pipeline. This thoroughness leaves little doubt on the empirical effectiveness of the method, especially across multiple languages.

- The work justifies the method well and makes it relatively easy to follow what is being done and why. This leads to a paper that reads easily and doesn't introduce unnecessary complexity in it's methods description or in it's analysis.

**Reasons To Reject:**

- It would have been nice to see more truly low-resource languages in the evaluation. The authors cite a lack of evaluation resources as the reason for this in limitations, but the MASSIVE dataset (released in June 2022 on Arxiv, though only at ACL 2023 in official publication) would have been a prime candidate to up the linguistic diversity of the experiments. Whether or not that work counts as contemporaneous depends on one's opinion on Arxiv vs. peer reviewed venues though, so not a major critique especially since it's acknowledged in limitations.

- While the transfer free framing of the method is novel, it seems heavily related to low resource scenarios in general. Ideally, the work would have also compared to data efficient learning techniques such as IA3 rather than only using full-finetuning as a baseline. Even if not full low-resource learning techniques, simple baselines of regularization to avoid overfitting on small data quantities would be great context for the technique. Without this, the work is slightly less sound since it's not clear how it compares to more established methods for low-data settings.

**Reproducibility:**

5: Could easily reproduce the results.

**Reviewer Confidence:**

5: Positive that my evaluation is correct. I read the paper very carefully and I am very familiar with related work.

**Typos Grammar Style And Presentation Improvements:**

Line 280: Introduce variable K before using it for "2K". Initially, this read confusingly as 2,000 since K hadn't been defined before use.

Table 2: Can you label all significant results with an asterisk or a sword to avoid needing the appendix to confirm significance. Full STDev could be left to the appendix, but marking whether a particular P value is reached in the main table would be useful.

The references to "Step 1" vs. "Stage 1" get slightly confusing and I found myself having to go back to remind myself which was a step and which was a stage. It might be easier to read if you use a notation that distinguishes granularity such as renaming Step 1 and 2 to "Stage 2.A and 2.B".

In between transfer-free and full transfer, it seems like there are cases where semantically related classes from one labeling scheme to another. The contrastive learning technique here seems like it would be useful pretraining in those cases. It seems worth noting that, especially since the full transfer-free setting is relatively extreme.

---

> ### Author Rebuttal · Authors · 2023-08-29
>
> Thank you for a really thorough review and thoughtful comments and suggestions! Please find the responses to the questions below. If given the opportunity, we will also address them in the camera-ready version of the paper.
>
> With regards to Reason to Reject 1 (about the low-resource languages), at the time of the experimentation for the paper the dialogue NLU datasets with low-resource languages (including MASSIVE) were not available. At the same time, as we were interested in the performance of TWOS in low-resource languages, we conducted a preliminary experiment for Yoruba (yor) and Luo (luo) languages on a related task of Named Entity Recognition (NER). The choice of the task was motivated by two reasons: a) similarly to slot labelling, NER is a sequence labeling task; b) there is a dataset built specifically for NER for low-resource African languages. The experiments were conducted with 100 training examples, using XLM-Roberta as a sentence encoder. TWOSL has brought considerable improvements for the languages, namely, 14.46 to 45.38 F-1 for yor and 16.56 to 36.46 F-1 for luo. These results prove the effectiveness of the method for low-resource languages as well. We will make sure to include these results into the camera-ready version if given the opportunity, and we also plan to run TWOSL on a representative subset of low-resource languages sampled from the MASSIVE dataset.
>
> With regards to Reason to Reject 2 (applying the techniques such as IA3), we thank the reviewer for a great suggestion! We did not include the PEFT results as the work focuses more on sample efficiency and not parameter efficiency for which methods like IA3 were proposed. At the same time, the training of the fully tuned model was regularised with more standard techniques such as early stopping and by using Adam optimiser with reducing learning rate. TWOSL can also work well with PEFT techniques such as IA3 and adapters, and this is definitely something to add to enrich the discussion in the paper.
>
> With regards to the questions:
>
> * A: Thanks for the question! We fully agree that more specialised, language-/language-family specific sentence encoders could also be applied in the setup instead of multilingual encoders. In our experimental setup we opted for the universality of the underlying models, i.e., making sure that we apply the same setup to all languages while not every language in our experiments had a suitable sentence encoder openly available. We will make sure to add this point to the potential future work directions in the final version of the paper - the main benefits of TWOSL remain in cases when the multilingual encoder is replaced with a monolingual language-specialised one.
>
> * B: In the experiments on MultiATIS++ (FIg. 3), we experiment with up to 800 in-language training examples available. These experiments show that the performance improvements scale with the number of training examples, although the gap between full finetuning and TWOSL gradually decreases as the number of training examples increases. We also ran a small experiment where we rely on large transfer data from English (with the target languages being German and French) - the results are reported in Table 4 of the main paper. It would be interesting to extend this set of experiments to more (lower-resource) target languages as well in future research.
>
> * C: In this work we focused on single-level slot annotations, while TWOSL can be extended to nested slot annotations by training another slot classifier on top of first level slot annotators. We see this as a potential extension of the method and will make sure to point to it as a potential future direction for the work - thanks for the insightful suggestion!
>
> * D:  In terms of methodology, it can be applied on top of tokenised as well as word-separated inputs. However, tokenized inputs would need alignment between predicted labels and the annotations and we opted to avoid this additional level of noise and discrepancy between TWOSL and baselines.
>
> Thanks a lot for the ideas on improvements for the notations! We will make sure to change the notations in the paper accordingly!

---

### Official Review · Reviewer_qbcu · 2023-08-05

**Soundness:** 4

**Excitement:**

4: Strong: This paper deepens the understanding of some phenomenon or lowers the barriers to an existing research direction.

**Paper Topic And Main Contributions:**

In this paper, the authors aim to address the inherent data scarcity issue on the new language-domain-task scenario of the slot labeling task.
The authors propose a two-stage slot labeling method to make the model (trained on English data) transfer-free in new scenarios.
By the proposed method, the model could perform well on few-shot setups.

**Reasons To Accept:**

The proposed method utilizes contrastive learning to bring the gap of the slot in different language-domain-task scenarios, and yields significant improvements in low-resource language scenarios, where only 50,100, or 200 training examples are in the training set.
Further, the proposed contrastive learning method could effectively distinguish different slot types.
Besides, the authors also provide detailed analyses of the proposed method in terms of the robustness and generalization, in the low-resource language scenarios.

**Reasons To Reject:**

1. Lack of the corresponding formulas of the contrastive learning objection, which makes it indirect to understand the content.
2. Lack of evaluation on low-resource domain scenarios.
3. Lack of solid comparison with other related methods.

**Reproducibility:**

4: Could mostly reproduce the results, but there may be some variation because of sample variance or minor variations in their interpretation of the protocol or method.

**Reviewer Confidence:**

4: Quite sure. I tried to check the important points carefully. It's unlikely, though conceivable, that I missed something that should affect my ratings.

---

> ### Author Rebuttal · Authors · 2023-08-29
>
> We are grateful to the reviewer for the insightful comments and suggestions and will make sure to address them in the camera-ready version of the paper!
>
> Regarding the Reason-to-Reject 1, we will make sure to spell out the exact formulas for the contrastive loss in section 3.1 in the camera-ready version of the paper. At the same time, we would like to note that the process of contrastive learning is described in detail in section 3.1 and the loss more specifically is described in lines LL 288-302.
>
> Regarding the Reason-to-Reject 2, the focus of the paper is on few-shot multilingual slot labelling. We agree with the reviewer that TWOSL could be applied in few-shot cross-domain scenarios as well and see it one as one of the directions for future work (as mentioned in LL.619-620). Given the additional page in the camera-ready version, we will gladly expand the description of potential application of TWOSL in multi-domain setups.
>
> Regarding the Reason-to-Reject 3, We aimed to run a comprehensive comparison of TWOSL with the standard and representative baselines which were relevant and could be reasonably applied in the transfer-free setup (and with different amounts of training data).  We would be happy to include comparison to additional  baselines, given the additional space in the camera-ready version of the paper.

---

### Meta-Review · Area_Chair_gp8X · 2023-09-19

**Recommendation:** 4

**Metareview:**

The paper proposes a two-stage contrastive learning based slot labeling approach to solve the language-domain-task adaptation challenge   for task-oriented dialogue system development. Reviewers agree that the work is well-motivated and achieves significant improvements in low-resource language scenarios. Several questions regarding the evaluation on truly low-resource languages, and comparison with seq2seq model were sufficiently addressed during rebuttal by authors.

---

### Decision · Program_Chairs · 2023-10-07

**Decision:**

Accept-Main

**Comment:**

The paper proposes a two-stage contrastive learning based slot labeling approach to solve the language-domain-task adaptation challenge   for task-oriented dialogue system development. Reviewers agree that the work is well-motivated and achieves significant improvements in low-resource language scenarios. Several questions regarding the evaluation on truly low-resource languages, and comparison with seq2seq model were sufficiently addressed during rebuttal by authors.